# Karst-environments of the southeastern Yucatan Peninsula: Hotspots for modern freshwater microbialites

Alfredo Yanez-Montalvo[1][¤a], Bernardo Águila[1][¤b], Arit S. de León-Lorenzana[1][¤c], Arturo Bayona[2], Nuria Torrescano-Valle[3], Pavel Popoca[1], Luisa I. Falcón [1]*

1 Instituto de Ecología, Unidad Mérida, Universidad Nacional Autónoma de México, Ucú, Yucatán, México, 2 Instituto Tecnológico Superior de Felipe Carrillo Puerto, Carretera a Vigía Chico Kilometro 1.5, Centro, Felipe Carrillo Puerto, Quintana Roo, México, 3 El Colegio de la Frontera Sur, Unidad Chetumal, A. del Centenario km 5.5, Chetumal, Quintana Roo, México

¤a Facultad de Ciencias, Universidad Autónoma de Baja California, Carretera Transpeninsular, Fraccionamiento Playitas, Ensenada, Baja California, México
¤b Instituto de Biología, Universidad Nacional Autónoma de México, Tercer Circuito s/n, Ciudad Universitaria, Coyoacán, Mexico City, Mexico
¤c CONAHCYT- Universidad Intercultural Maya de Quintana Roo, Centro de Innovación para el Desarrollo Apícola Sustentable. Carretera Federal Muna Felipe Carrillo Puerto Km 137 s/n, 77870, La Presumida, José María Morelos, Quintana Roo, México
* falcon@ecologia.unam.mx

## Abstract

Modern microbialites are sedimentary structures that offer a window into Earth's geologic history and the intricate interplay between geology and microorganisms. Microbialites are formed by the interaction between microbial communities and the environment leading to mineral precipitation. This study provides a comprehensive analysis of the bacterial and archaeal composition (using the V4 region of the 16S rRNA), along with mineralogy, geochemistry, and hydrogeochemical characterizations of microbialites of five aquatic systems (Bacalar, Muyil, Chichancanab, Azul and Cenote Azul) in southeastern Yucatan Peninsula, México. Dominant taxa were distributed within Pseudomonadota, Cyanobacteriota, Bacillota, Bacteroidota, Chloroflexota, and Planctomycetota, while NB1-j, Myxoscoccota, Verrucomicrobiota, Acidobacteriota, and Crenarchaeota (Archaea) were less abundant. Microbialites from Cenote Azul, a deep sinkhole, were the most different and biodiverse. Notably, potential new families of Cyanobacteriota were observed in all microbialite sites. The primary mineral constituents in microbialites were calcite, magnesian calcite, and gypsum. Hydrogeochemical conditions differed among sites despite their hydrological connectivity. Overall, the karstic ecosystem, hydrogeochemical conditions, tropical climate, and shallow coastal landscapes have favored the occurrence of microbialites in the Yucatan Peninsula, a hotspot region for the formation of these communities. However, their safeguarding becomes crucial, emphasizing the urgency of our role in environmental conservation, in the face of challenging conditions associated with climate change and increased anthropogenic activities detrimental to the environment.

**Data availability statement:** The data supporting the findings of this study are available in the NCBI GenBank database under BioProject PRJNA1123232 (https://www.ncbi.nlm.nih.gov/bioproject/PRJNA1123232).

**Funding:** This work had funding from UNAM, PAPIIT (IN204224) to LIF. AY and BA hold postdoc fellowships from CONAHCyT and UNAM-PAPIIT, respectively.

**Competing interests:** The authors have declared that no competing interests exist.

## Introduction

The Yucatan Peninsula is a karstic platform with various landscapes characterized by a distinctive hydrology and geology, including caves, springs, lagoons (in this study we will use the term "lake"), rivers and sinkholes (locally called 'cenotes' from the Mayan word '*dzonot*'), formed from the dissolution of carbonate rocks [1]. Karst ecosystems constitute approximately 15% of the Earth's surface and provide habitat for nearly 20% of the global population [2]. Given their close association with freshwater dynamics, karstic ecosystems are pivotal in sustaining life [3].

The Yucatan Peninsula is composed of three Mexican states (Campeche, Quintana Roo and Yucatan), portions of northern Belize and Guatemala, forming an approximate area of 450,000 km$^2$ that harbors a vast terrestrial and aquatic biodiversity [4]. The state of Quintana Roo is home to the largest Maya-speaking community in the Yucatan Peninsula. Here, the primary source of income is intensive conventional tourism, which receives over 16 million tourists a year, concentrated in Cancun, Playa del Carmen and Tulum [5]. Intense tourism, population growth without development plans, and conventional agriculture are the primary drivers of land use change, compromising water quality [6]. Quintana Roo harbors several distinct ecosystems, including the Mesoamerican Reef System (SAM) in the Caribbean Sea [7], the most extensive underwater cave systems [8], the deepest known blue hole [9], the Sian Ka'an Biosphere Reserve (a UNESCO site) home to endangered species of flora and fauna, such as jaguars, tapirs, and mangrove forests [10] and the largest extant freshwater microbialite reef which occurs in Bacalar lake [11,12]. Given its karstic origin, the region presents a complex subsurface and groundwater hydrological system with oligotrophic characteristics, limited in phosphorus and nitrogen, with ion saturation of bicarbonates, calcium, and magnesium [13,14]. Southeastern Yucatan Peninsula is characterized by a series of aquatic and terrestrial ecosystems interconnected in the surface and groundwater, forming the Transverse Coastal Corridor [15].

Modern microbialites are organic-sedimentary structures formed by microbial communities (mainly Bacteria and Archaea) coexisting with a high biodiversity of eukaryotes and viruses [16–18]. Microbialites usually develop in environments saturated in bicarbonate, calcium, and magnesium, among other ions [19]. Extracellular polymeric substances (EPS) secreted mainly by Cyanobacteriota are fundamental in alkaline environment formation, cation binding, and release, allowing the formation of carbonate precipitates [17,20]. Additionally, high metabolic diversity, including photosynthesis, sulfate reduction, denitrification, ammonification, methane oxidation, and ureolysis, favors carbonate mineral precipitation [21,22]. Microbialites are ecosystems that hold high ecological and evolutionary importance globally but are seldom recognized beyond the scientific community, and a proper recognition of their ecological and cultural value by decision-makers to implement conservation policies for these microbial communities has yet to be achieved. Currently, few examples exist of protection for the conservation and visibility of microbialites, such as the Hamelin Pool Marine Nature Reserve in Australia and the Stromatolite Park in Tierra del Fuego, Chile. The city of Bacalar in Quintana Roo, Mexico, celebrates "Stromatolite Day" on July 15.

Since the Yucatan Peninsula aquifer has the conditions to harbor microbialite formations, here we aim to evaluate the hypothesis of high connectivity between aquatic systems in karst environments and to elucidate whether these sites share a common microbial community associated with microbialites. In this study, we provide a comprehensive hydro-geochemical profile of five microbialite sites and analyze their mineralogy, elemental geochemistry, bacterial and archaeal composition.

## Materials and methods

### Study site and sample collection

The Yucatan Peninsula is a unique bioregion, characterized by evaporitic rocks and high ecological connectivity through surface and groundwater flow. It has been a significant area of study for microbialites, with several sites reported, including Chetumal Bay [23], Muyil lake, Sian Ka'an Biosphere Reserve [24], Bacalar lake [12,24–26], Cenote Azul [27,28], Chichancanab and Azul lakes (reported here) (Fig 1).

In this study, we meticulously focused on five sites that harbor microbialites, including lakes (Azul, Bacalar, Chichancanab and Muyil) and sinkhole (Cenote Azul). For each site, we collected 5 samples of approximately 2.5 cm in diameter

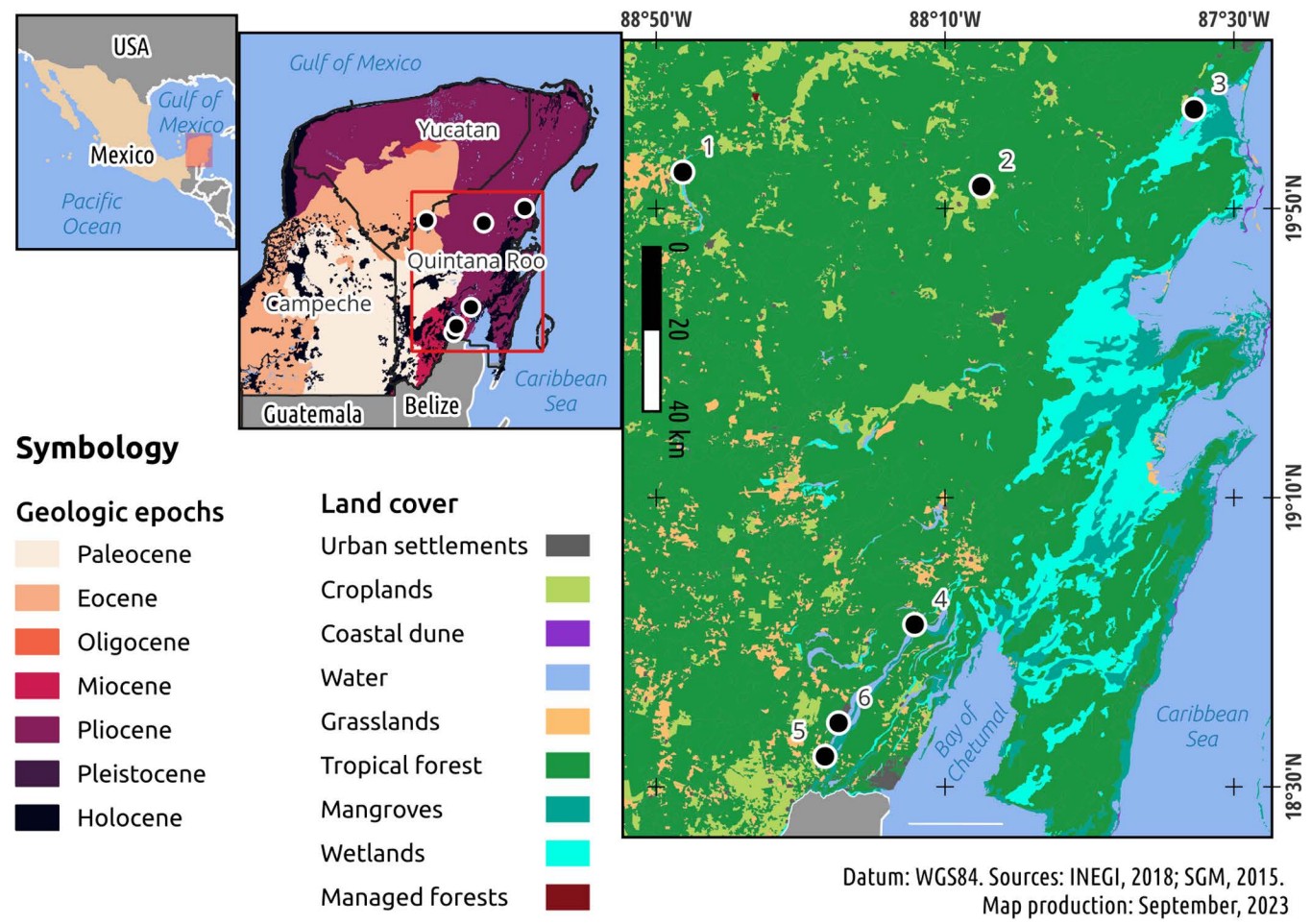

**Fig 1. Distribution map of the sampling sites along the Yucatan Peninsula.** General information on the coordinates, geology and land use change of lakes with microbialites (S1 Table).

from 10 microbialites. The samples were collected with gloves and sterile material, placed in cryotubes, and stored and frozen in a liquid nitrogen tank until processing in the laboratory. All microbialite samples were collected under collection permit SPARN/DGVS/00223/25 granted by SEMARNAT, Mexico (LIF). Field studies were conducted carefully and did not include protected or endangered species. Sampling was done in the winter of 2019 (Bacalar and Cenote Azul) and spring of 2020 (Muyil, Chichankanaab and Azul).

Bacalar lake develops along a geological fault and is the largest superficial freshwater lake in the Yucatan Peninsula [29]. It is home to the largest freshwater microbialite-reef in the world, which has been spatially characterized in two bioregions distributed in the North and South of the lake, which have been described extensively by Yanez-Montalvo et al. [12].

Cenote Azul is an open karst sinkhole separated by less than 200 m from Bacalar lake, yet they are not connected hydrologically. It has a depth of approximately 65 m and a diameter of 200 m, a microbialite reef develops along its periphery and depth [27,30].

Chichancanab lake (small-sea in Maya) is a freshwater system that spans over 20 kilometers. Its hydrological, geological, and chemical characteristics are distinct. Originating approximately 8,200 years ago [31], early Holocene sediments reveal the presence of the benthic foraminifera *Ammonia beccarii* skeletons, indicating salinities ranging between 13 and 30 g/L which are favorable conditions for the organism's reproduction. Around 7,200 years ago, *A. beccarii* vanished from the sediments, suggesting a gradual decline in salinity. This shift is attributed to potentially more humid climates in the Middle Holocene, leading to a transition towards freshwater conditions [31]. The immediate environs of these water bodies encompass approximately 1400 hectares of periodically flooded areas, featuring grassland savannas, buttonwood mangrove (*Conocarpus erectus*), navajuela (*Cladium jamaiscensis*), and cattail (*Typha latifolia*).

Muyil lake is inside the Sian Ka'an Biosphere Reserve and is part of the Muyil-Chunyaxche Lake System. It is one of the largest superficial freshwater systems in the Yucatan Peninsula, along with Bacalar and Chichancanab lakes. The lake is within a natural protected area decreed in 1986 as Sian Ka'an Complex and is made up of the Sian Ka'an Biosphere Reserve, Uaymil area of flora and fauna protection, and Sian Ka'an Reefs Biosphere Reserve. It is included in the International Network of Biosphere Reserves and has been a World Heritage Site since 1987. The surrounding area has medium-sized rainforest, mangroves and coastal wetlands. Currently, there are low-impact ecotourism activities that do not impact the microbialite formations that exist in its waters.

Azul lake spans approximately 1 km in length and 800m at its widest point. Azul encompasses a system of four sinkholes in its vicinity, lacks inlet or outlet channels, and relies on groundwater for sustenance. The lake shares a similar floristic and faunal stratum with Muyil-Chunyaxché and Chichancanab, featuring medium-sized rainforest and wetlands. Economic activities in the region center around low-impact community ecotourism, offering opportunities for canoeing and hiking around the site. Due to its remote location, the lake's natural attributes remain exceptionally well-preserved. Notably, Azul lake is the most recent site where microbialites were discovered, underscoring its ecological importance and the need for continued preservation efforts.

## Hydrogeochemistry, mineralogical and geochemical characterization of microbialites

Hydrogeochemical analyses were performed at the Institute of Geology (UNAM, LANGEM-PT-LCL). Ion chromatography with conductometric detection in Waters 1525 binary HPLC pumps, autosampler (model No. 717), and conductivity detector (model No. 432) were used. An IC-Pak Anion HR stationary column (Waters) with 5 µm amino packing was used. The sample volume used in the chromatograph was 10 µl. For anion characterization, a mobile phase consisting of a mixture of acetonyl, butanol and a solution of sodium gluconate/borate in water was used with a flow rate of one mL/min. For cations, the mobile phase consisted of dipicolinic acid and nitric acid (1.7 mM each) in water with a flow rate of 0.9 mL/min.

Mineralogical analyses of microbialites were carried out at the X-Ray Diffraction Laboratory of the Institute of Geology, UNAM. The samples were cold dried (10 °C), ground in an agate mortar, and sieved (mesh < 75 µm). The equipment employed a 2θ angular range from 5° to 80° in a step scanner with a "step scan" of 0.003° and an integration time of

40 sec per step, using double-sided aluminum supports (non-oriented fractions). A diffractometer (Empyrean) equipped with a Ni filter, a monochromator, a thin copper tube focus, and a PIXcel3D detector was used to obtain the diffractogram. The diffraction patterns were analyzed with HighScore software (version 4.5) with reference patterns from the ICDDPDF-2 and ICSD databases.

Geochemical analyses were performed at the Paleoecology laboratory of ECOSUR, Chetumal Unit. A fraction of the samples was dried (40 ºC) and ground in an agate mortar, and a Thermo Scientific Niton XL3T X-ray fluorescence analyzer (XRF) was used to determine the geochemistry. The measured elements were corrected through linear equations according to the method of Quiroz-Jiménez and Roy [32] and Roy et al. [33]. The fitting equations were developed using 15 reference materials (IGLsy-1GEOL UNAM, IGLa-1 GEOL UNAM, IGLd-1 GEOL-UNAM, IGLgb-3 GEOL-UNAM, IGLs-1 GEOL-UNAM, IGsy-4 GEOL UNAM, IGLsy-2 GEOL UNAM, IGLc-1 GEOL UNAM, as well as Es-2 Black Shale Estonia 2000, Es-4 Dolostone EST 1A, NIST 2709a PP 180–649, NIST 2702, CCRMP TILL-4PP 180–646 and QC USGS SAR-M 180–673, RCRApp).

Principal component analysis (PCA) was performed using an euclidean distance matrix to understand the effect of environmental parameters and the elemental chemical composition of microbialites from Quintana Roo. Mineralogy and geochemistry measurements were used to build a dendrogram with the hclust hierarchical clustering function, using Ward's method.

## Microbialites DNA extraction and 16SrRNA gene amplification

Total genomic DNA was extracted from microbialite samples (0.25g) (5 subsamples x 10 microbialites/locality) using the DNeasy PowerSoil® Kit (Qiagen), following the manufacturer's instructions. An amplicon library of the V4 hypervariable region of the small subunit of the 16S rRNA gene was prepared for each sample with the 515F/806R primers, following an established protocol [34]. Triplicate PCRs followed the program: 98°C for 30s followed by 35 cycles of 95°C for 30s, 52°C for 40s, and 72°C for 90s, and a final elongation step of 12min at 72°C, then kept at 4°C. PCR products were visualized on 1% agarose gels and purified using Ampliclean carboxyl-coated magnetic beads (NimaGen, NDL). The purified amplicon libraries were quantified (QUBIT fluorometer, Promega, USA) and sequenced (20 ng/μl sample) on an Illumina Miseq 2×300 platform (Yale Center for Genome Analysis, CT, USA).

## 16S rRNA amplicon processing

Sequence data processed using the Quantitative Insights into Microbial Ecology 2 (QIIME 2, v. 2022.2) pipeline [35], were deposited in the GenBank under BioProject PRJNA1123232. Reads (forward and reverse) were trimmed in position 20 from the 5' end and truncated to a length of 220. DADA2 pipeline was used for quality filtering, denoising, paired-end merging, and assigned into (amplicon sequence variant) [36]. The alignment was used to construct a tree and calculate phylogenetic relations (FastTree) [37]. SILVA (v.138) taxonomic classification was used as a reference database [38], were aligned using MAFFT [39], and highly variable regions were masked from the alignment. Subsequently, diversity analyses and data visualization were performed in the R environment (v 4.0.3). Finally, the dataset was prepared by removing sequences representing less than 1,000 reads, singletons, chloroplasts, mitochondria, and sequences not classified at the Phylum level.

## Microbial composition, community structure and statistical analysis

Prokaryotic community analysis was performed using the R environment, the following packages were used in Phyloseq [40], the vegan package [41], the ampvis2 package [42], and the package collection in tidyverse [43]. Alpha diversity indices were explored using the Chao1, Shannon, and InvSimpons metrics. Alpha diversity comparisons were performed using Kruskal-Wallis non-parametric test. The structure and taxonomic composition associated with the Quintana Roo microbialites was explored by bar chart analysis at the Phylum level (<2%).

Beta diversity was represented with UniFrac (weighted and unweighted) distance [44] and represented in a Non-metric Multidimensional Scaling analysis (NMDS). We evaluated the taxonomic composition of microbialites across all sites. Permutational multivariate analysis of variance (PERMANOVA) was executed with the adonis2 function, and analysis of similarity (ANOSIM) [45]; both analyses were conducted within the vegan package, incorporating UniFrac distances (weighted and unweighted) along with 999 permutations to ensure a robust statistical assessment. Pairwise PERMANOVAs between sites with microbialites were conducted with 999 permutations using the pairwise. perm.manova function in the RVAideMemoire package [46] with a Benjamini–Hochberg adjustment of $p$-values. The difference in dispersions between groups was evaluated with the betadisper test ($p$-value threshold of 0.05). The comparison between the distribution of ASV's was represented with an UpSet plot [47]. BDTotal was partitioned into Local Contribution to β-diversity (LCBD) and Species Contribution to β-diversity (SCBD), following Legendre and De Cáceres [48]. β-total diversity (BDtotal), LCBD, and SCBD were calculated by transforming the species abundance matrix (method = Hellinger) with 999 permutations to obtain the total sum of squares (SStotal). We also applied p-adjustment using the 'holm' method to correct for multiple comparisons in LCBD. The 15 most abundant genera were obtained for each microbialite site.

## Results

### Mineralogy and geochemical characterization of water column and microbialites

Water column physicochemical parameters and hydrogeochemistry varied among sites (Table 1 and 2). pH ranged slightly (7.8–8.1); bicarbonates ranged between 160–215 mg/l, and Azul lake and Cenote Azul had the lowest and highest concentrations, respectively. The highest sulfate concentrations were found in Chichancanab lake (2804 mg/l), while Muyil lake had the lowest values (50.66 mg/l). Chlorides showed the highest values in the three sites located north of

**Table 1. Physicochemical parameters and hydrogeochemistry (anions) of the water surrounding the microbialites from the six sampling locations.**

| Location | Conductivity | pH | Sulfate- | Bicarbonate- | Chlorine- |
|---|---|---|---|---|---|
| | mS/cm | | mg/l | | |
| Chichancanab lake | 2.1 | 7.8 | 2804.01 | 190.59 | 242 |
| Azul lake | NA | 8 | 202.37 | 160.27 | 287 |
| Muyil lake | 1.4 | 8.1 | 50.66 | 204.62 | 325 |
| Bacalar lake (North) | 3.5 | 7.4 | 1159.17 | 125.49 | 161.08 |
| Bacalar lake (South) | 2.1 | 7.6 | 1137.62 | 177.39 | 53.68 |
| Cenote Azul | 2.3 | 8 | 1235.27 | 215.6 | 62.12 |

**Table 2. Hydrogeochemistry (cations) of the water surrounding the microbialites from the six sampling locations.**

| Location | NOx* | Calcium | Ammonium | Magnesium | Sodium |
|---|---|---|---|---|---|
| | mg/l | | | | |
| Chichancanab lake | 0.589 | 700.02 | 0 | 218 | 192 |
| Azul lake | 1.65 | 67.05 | 1 | 55 | 146 |
| Muyil lake | 21.2 | 64.65 | 0.87 | 40 | 164 |
| Bacalar lake (North) | 5.94 | 370.69 | 2.62 | 94 | 116 |
| Bacalar lake (South) | 12.52 | 358.05 | 7.4 | 173.85 | 51 |
| Cenote Azul | 7.03 | 424.07 | 0 | 86.05 | 35 |

*NOx (sum of nitrates and nitrites)

the Yucatan Peninsula: Azul lake, Chichancanab lake, and Muyil lake, with the latter having the highest value (325 mg/l). NOx (sum of nitrites and nitrates) was lowest in Chichancanab lake (0.6 mg/l) and highest in Bacalar lake (South) and Muyil lake, with 12.52 and 21.19 mg/l, respectively. Ammonium was highest in Bacalar lake (North and South), with 2.62 and 7.4 mg/l, respectively. Sodium was highest in Chichancanab lake (192 mg/l). Magnesium and calcium were highest in Chichancanab lake, with 218 and 700 mg/l, respectively.

Determinations of mineral composition by X-Ray Diffraction revealed a distinct compositional pattern within the microbialite formations. Specifically, the southern region encompassing Bacalar lake (North and South), and Cenote Azul sites exhibited elevated Calcite content, ranging from 94% to 98.6%. Additionally, trace amounts of other minerals were identified, including Siderite, Kieserite, Thenardite, Hematite, Quartz, Magnesite, and Rhodochrosite. In the northern locations, Chinchankanab lake and Azul lake had percentages higher than 98% of Magnesian Calcite. Muyil lake was composed mainly of Magnesian Calcite (61%) and gypsum (39%) (Table 3).

The geochemical composition of all sites was examined by X-ray fluorescence. Specifically, the presence of Niobium was solely identified in Chichancanab lake and Cenote Azul. The highest level of Rubidium, reaching 18.14 ppm, was observed in Chichancanab lake. Iron exhibited higher concentrations in the three lakes in the northern part of the study area, ranging from 0.47 to 0.48 ppm. Strontium was detected in significantly higher concentrations within the microbialites of Chichancanab lake, reaching 2563 ppm. Calcium was the most homogeneous element in the composition of the microbialites (Table 4).

Principal Component Analysis (PCA) provided insights into the hydrogeochemistry of the water surrounding microbialite growth (Fig 2a). The spatial arrangement of sites in the quadrat aligns with their geographic locations, as depicted in Fig 1. Specifically, in the southern sites (Bacalar lake North, Bacalar lake South, and Cenote Azul). On the other hand, Bacalar lake (North) shows a higher correlation with conductivity, while Bacalar lake (South) exhibits a stronger association with

**Table 3. Mineral composition by X-Ray Diffraction of microbialites.**

| Location | Calcite | Magnesian calcite | Gypsum | Aragonite | Others* |
|---|---|---|---|---|---|
| | % | | | | |
| **Chichancanab lake** | 0 | 99.65 | 0 | 0.35 | 0 |
| **Azul lake** | 0 | 98.25 | 0 | 1.75 | 0 |
| **Muyil lake** | 61.15 | 0 | 38.85 | 0 | 0 |
| **Bacalar lake (North)** | 98.1 | 0 | 0 | 0 | 1.9 |
| **Bacalar lake (South)** | 98.6 | 0 | 0 | 0 | 1.4 |
| **Cenote Azul** | 94 | 1.25 | 0 | 0 | 4.75 |

*Others: Siderite, Kieserite, Thenardite, Hematite, Quartz, Magnesite, and Rhodochrosite.

**Table 4. Elemental chemistry (XRF) of the six locations with microbialites.**

| Location | Calcium | Rubidium | Iron | Niobium | Strontium |
|---|---|---|---|---|---|
| | % | ppm | | | |
| **Chichancanab lake** | 25.44 | 18.14 | 0.48 | 20.12 | 2563.05 |
| **Azul lake** | 31.52 | 16.02 | 0.47 | 0 | 2195.74 |
| **Muyil lake** | 35.47 | 14.94 | 0.47 | 0 | 1456.48 |
| **Bacalar lake (North)** | 36.02 | 13.88 | 0.014 | 0 | 2102.03 |
| **Bacalar lake (South)** | 37.05 | 8.53 | 0.016 | 0 | 2192.04 |
| **Cenote Azul** | 38.15 | 10.35 | 0.017 | 1.55 | 1747.05 |

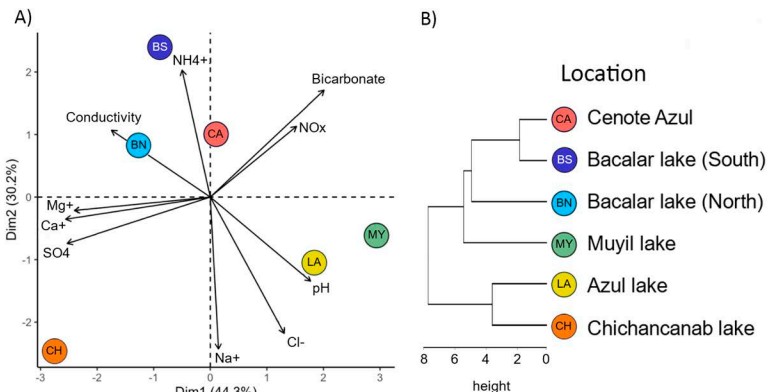

**Fig 2. (A) PCA of the hydrogeochemical variables of the surrounding water of microbialites.** (B) Dendrogram on the mineralogy and geochemistry of microbialites.

ammonium. Northern sites, Azul lake and Muyil lake, show a more pronounced connection with pH, while Chichancanab lake is correlated with sulfate, calcium, and magnesium. The dendrogram, constructed based on the mineralogy and geochemistry database, reveals three distinct groups: the first comprising Chichancanab lake and Azul lake, the second solely encompassing Muyil lake, and the third formed by Bacalar lake (North and South), and Cenote Azul (Fig 2b).

## Bacteria and Archaea composition based on 16S rRNA

A total of 21,846,988 16S rRNA gene sequences were obtained. After quality filtering and chimera removal, the remaining sequences were 14,826,500. The dataset was exported to the R environment, and a phyloseq file was created and filtered to obtain a total of 41,363 ASVs 39,725 of which are Bacterial and 1,638 Archaeal.

## Diversity metrics, Microbial structure, and composition

Alpha diversity estimators showed the particularity of the microbial diversity of microbialites with indices including Chao1, Shannon, and InvSimpson (Fig 3). For all Alpha diversity values, the diversity associated with the microbialites from Cenote Azul was the highest, richness (Chao1 = 1720), Shannon-Wiener index ($H'$ = 6.3) and equitability (D = 125) (the values are shown in S2 Table). Differences between pairwise analyzed Alpha diversity metrics are shown in the S2 Table. The Bacalar lake (South) microbialites exhibit a distinct trend compared to other microbialite reservoirs, with all metric values registering as the lowest among the sites analyzed.

The microbialites of Quintana Roo were found to harbor highly diverse microbial communities (Fig 4). At the domain level, Cenote Azul had 89.1% Bacteria and 10.9% Archaea, whereas all other lakes had 99.8–99.7% Bacteria and 0.3–1.2% Archaea. In total, the taxonomic analysis identified 81 phyla in Quintana Roo´s microbialites. The most ubiquitous phyla were Pseudomonadota, Cyanobacteriota, Bacillota, Bacteroidota, Chloroflexota, Planctomycetota, NB1-j, Myxococcota, Verrucomicrobiota, and Acidobacteriota. Among Archaea, the phylum Crenarchaeota accounted for more than 50% of the diversity, followed by Nanoarchaeota, Asgardarchaeota, Thermoplasmatota, Euryarchaeota where the most abundant percentage at the family level was within Nitrosopumilaceae (S1 Fig).

Pseudomonadota was the most abundant bacterial phylum (with Alphaproteobacteria (8.2–29.8%), Gammaproteobacteria (7.3–26.6%), Cyanobacteriota (4–21.4%), Bacillota (0.8–15.5%), Chloroflexota (3.2–11.4%), Bacteriodota (2.2–12.8%), Planctomycetota (4.7–12.4%), Myxococcota (2.1–4.3%), NB1-j (1.7–5.6%), and Verrucomicrobiota (1.6–3.6%) classes representing up to 80% of the composition (Fig 4b). At the family level, the most abundant were Pseudomonadaceae (0–21.2%), an unknown Cyanobacteriota (1.1–9.3%), Hyphomonadaceae (0.8–8.0%), Microscillaceae (0.2–8.5%),

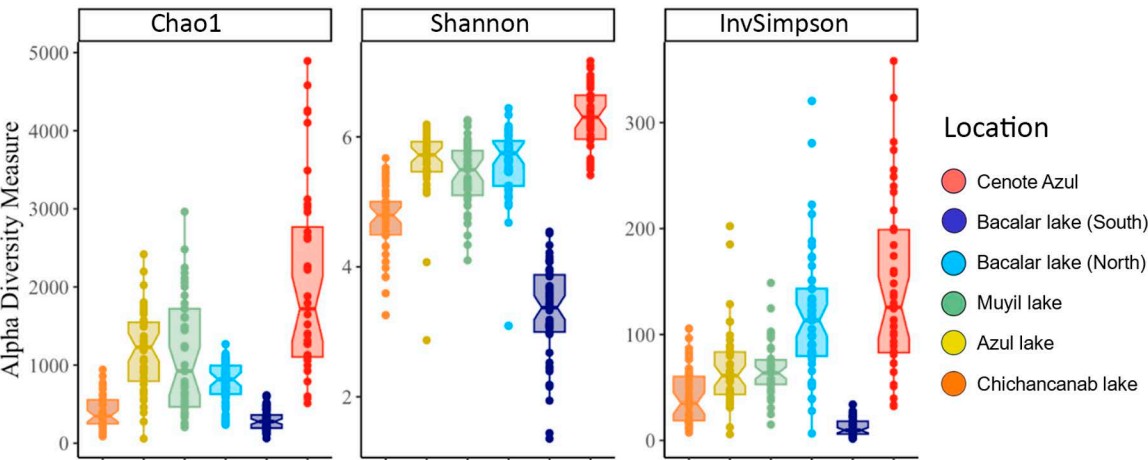

**Fig 3. Alpha diversity across sites with microbialites in Quintana Roo.** Three alpha diversity metrics Chao1, the Shannon and InvSimpson diversity indices, are shown as boxplots.

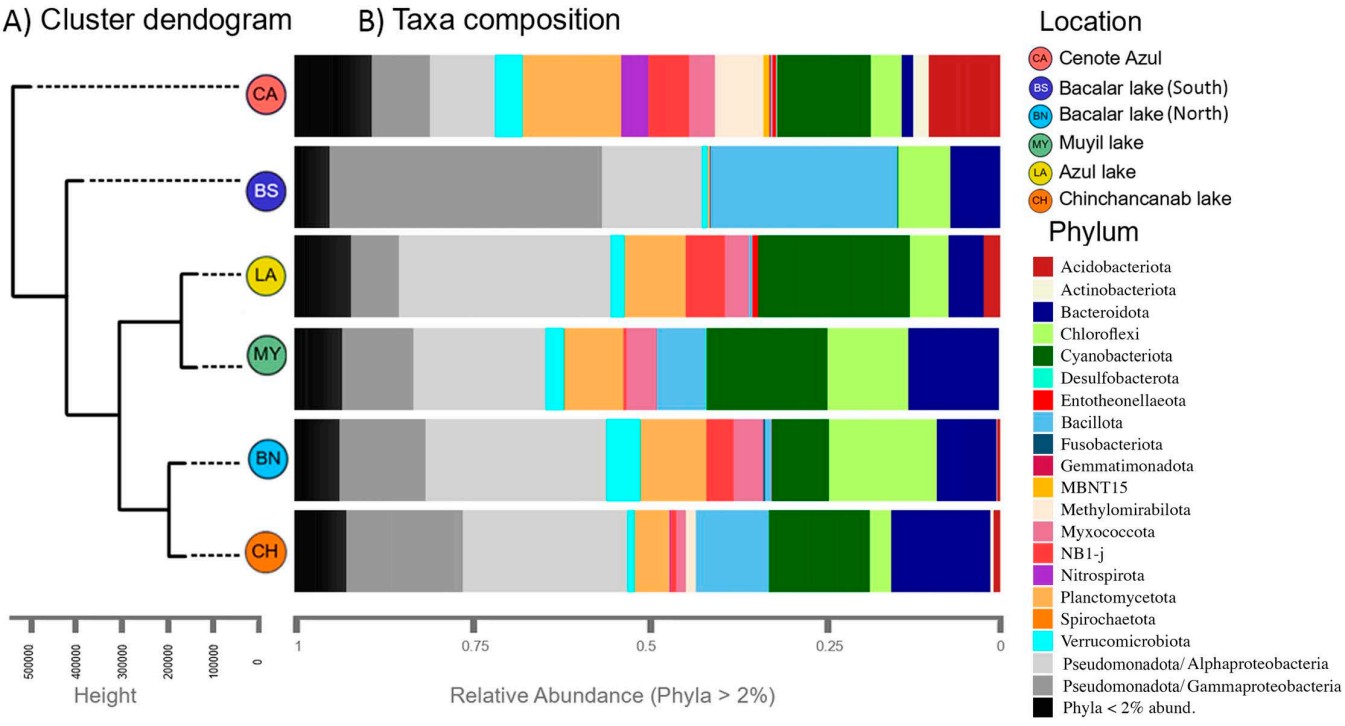

**Fig 4. Dissimilarity analysis and taxa composition.** (a) **Euclidean distance dendrogram (beta diversity) and** (b) **community composition at the phylum taxonomic level.**

A4b-Chloroflexi (0.5–5.9%), Bacillaceae (0.1–6.4%), and Nostocaceae (0.1–10.7%) (S2 Fig). Four microbialite groups were arranged according to similarity, revealing the intricate and complex nature of these communities.

Notable distinctions emerge between Cenote Azul and Bacalar lake (South), setting them apart from the other locations. Cenote Azul microbialites, sampled at depths of 10–30 m exhibit a higher diversity with relative abundances

surpassing 2% relative to other sites. In contrast, surface microbialites (1 m depth) obtained from the rest of the sampling sites share a more uniform composition, except for the Bacalar lake (South) site, which has a lower diversity (Fig 4). Nostocaceae and Microcystaceae are the families with the most dominant genera for shallow sites and for deep microbialites, respectively, and two unknown Cyanobacteriota families were identified within the most abundant groups (S2 Fig).

To analyze microbial beta-diversity, we realized a correlation matrix based on weighted and unweighted UniFrac distance matrix visualized employing ordination methods such as Non-metric Multidimensional Scaling (NMDS). Visual examination of the NMDS ordination revealed distinct separation in microbial communities associated with microbialites (Fig 5). The Cenote Azul group retained a tight cluster in both representations (weighted and unweighted UniFrac) (Fig 5). Notably all lakes, Chichancanab, Azul, Muyil, and Bacalar (North) clustered together except for Bacalar lake (South), the site with the lowest diversity. PERMANOVA tests demonstrated statistically significant differences between all locations, which was evident in terms of community similarity ($p$-value = 0.001), composition ($p$-value = 0.001), and homogeneity ($p$-value = 0.001) (S3 Table). Additionally, we compared the microbialites in two categories: shallow (Azul, Bacalar, Chichancanab and Muyil lakes) and deep microbialites (Cenote Azul) (S3 Fig).

According to the results of the ASVs clustering analysis, the shared and unique ASVs among the different microbialites were analyzed and plotted on an Upset plot (Fig 6). We observed that 244 ASVs are shared among the evaluated microbialites. Cenote Azul was the site with the highest number of associated ASVs (12,764), while Bacalar lake (South) was the site with the lowest number of ASVs (1689).

The LCBD and SCBD values describe the relative contributions of beta diversity at each local site and the relative contributions of the different ASVs (S4 and S5 Table). The BDTotal was 0.63, and the SSTotal was 3.19. LCBD values ranged from 0.13 to 0.35. Beta diversity assessed through the LCBD metric showed that turnover occurs at the Cenote Azul and Bacalar lake (South) sites ($p$-value < 0.05). The SCBD measurement showed that the ones that had the greatest contribution to changes related to Beta diversity were *Rhodomicrobium* (Alphaproteobacteria), *Rivularia* (Cyanobacteriota), *Pseudomonas* (Gammaproteobacteria), Unknown_Family (Cyanobacteriota) (the top 20 of the genera with the highest contribution is shown in S5 Table).

The 25 most abundant genera for the Quintana Roo microbialites are shown (Fig 7). Out of the most prevalent taxa within the six study sites, the genus A4b of Chloroflexota was higher or more prevalent in Bacalar lake (North) and Muyil

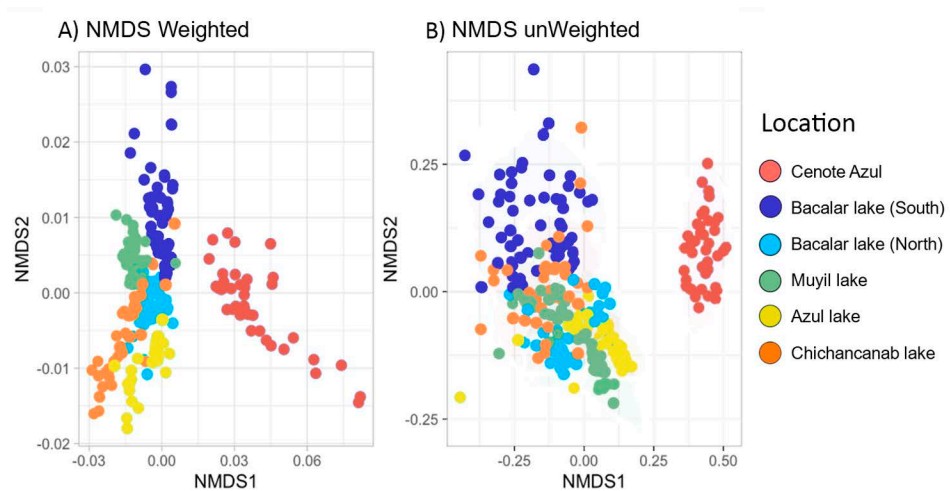

**Fig 5. NMDS plots of the microbial diversity showing UniFrac** (A) **distance between sites with microbialites,** (B) **sampling locations.** (A) showing weighted (a, stress: 0.114) and unweighted (b, stress:0.1) and Plots (B) showing weighted (a, stress: 0.114) and unweighted (b, stress:0.1).

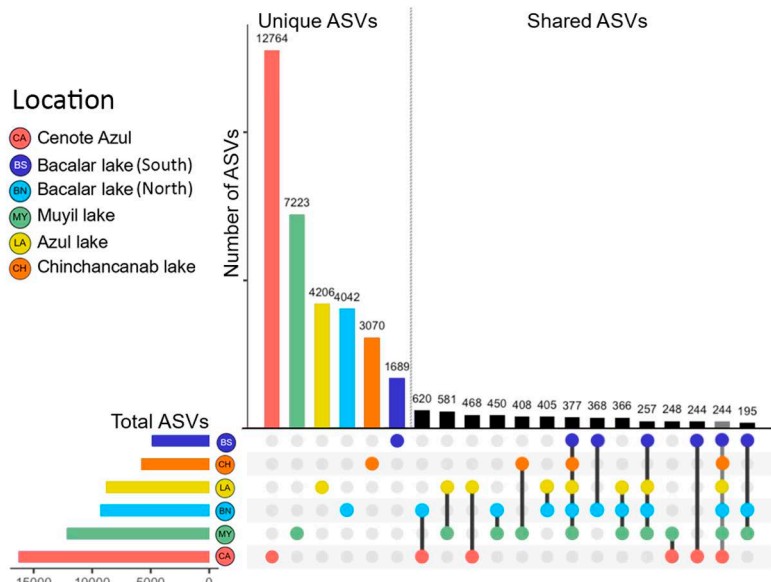

**Fig 6. Upset plot analysis of unique and shared microbialites sites in Quintana Roo.**

| | Bacalar lake (North) | Bacalar lake (South) | Cenote Azul | Chichancanab lake | Azul lake | Muyil lake |
|---|---|---|---|---|---|---|
| Pseudomonadota/Proteobacteria; uncultured | 6.8 | 0.7 | 3.6 | 2.3 | 11.6 | 5.2 |
| Pseudomonadota/ Proteobacteria; Pseudomonas | 0.4 | 25.8 | 0 | 4.5 | 0.1 | 1.5 |
| Chloroflexi; A4b | 8.5 | 2.6 | 0.6 | 1.5 | 2.1 | 5.9 |
| Planctomycetota; uncultured | 3 | 0.4 | 3.5 | 2.9 | 3.1 | 4.4 |
| NB1-j; NB1-j | 3.6 | 0.3 | 5.2 | 1.6 | 4.6 | 1.2 |
| Pseudomonadota/Proteobacteria; Rhodomicrobium | 0.8 | 0.3 | 0 | 11.2 | 1 | 2.1 |
| Chloroflexi; SBR1031 | 4.6 | 3.2 | 0.4 | 0.8 | 0.8 | 4 |
| Cyanobacteriota ; Rivularia_PCC-7116 | 1.3 | 0.2 | 0 | 0.3 | 8.7 | 2 |
| Bacillota/Firmicutes; Bacillus | 0.1 | 8.8 | 0 | 2.2 | 0 | 0.9 |
| Cyanobacteriota; uncultured | 1.6 | 0 | 0.5 | 2.7 | 3.2 | 2.5 |
| Bacteroidota; Microscillaceae | 0.9 | 2.3 | 0 | 1.1 | 0.5 | 5.5 |
| Cyanobacteriota ; Cyanobacteriales | 2.3 | 0.2 | 0 | 3.4 | 0.6 | 2.7 |
| Cyanobacteriota ; Unknown_Family | 0.7 | 0.1 | 0 | 1.8 | 0.6 | 5.8 |
| Pseudomonadota/ Proteobacteria; Hyphomicrobium | 1.6 | 0.3 | 0.2 | 1.1 | 3.9 | 1.7 |
| Cyanobacteriota; Microcystaceae | 0 | 0 | 2.2 | 0 | 0 | 0 |
| Crenarchaeota; Nitrosopumilaceae | 0.1 | 0.1 | 3.8 | 0.2 | 0.3 | 0 |
| Pseudomonadota/Proteobacteria; Aeromonas | 0.1 | 6.6 | 0 | 0.3 | 0 | 0.1 |
| Cyanobacteriota; Chloroplast | 0.9 | 0.1 | 0.7 | 2.2 | 0.2 | 1.3 |
| Bacteroidota; uncultured | 1.1 | 0.1 | 0.8 | 3.2 | 0.7 | 0.7 |
| Pseudomonadota/Proteobacteria; SWB02 | 2 | 0.1 | 0.7 | 0.7 | 1.7 | 1.4 |
| Pseudomonadota/Proteobacteria; Stenotrophomonas | 0.2 | 6 | 0 | 0.4 | 0 | 0.2 |
| Pseudomonadota/Proteobacteria; Brevundimonas | 0.1 | 6 | 0 | 0.4 | 0 | 0.2 |
| Planctomycetota; OM190 | 0.6 | 0 | 2.7 | 0.1 | 0.3 | 0.3 |
| Nitrospirota; Nitrospira | 0.2 | 0 | 2.6 | 0.3 | 0.2 | 0.2 |
| Planctomycetota; mle1-8 | 1.9 | 0.3 | 0.5 | 0.5 | 2.1 | 1 |

% Read Abundance
25
20
15
10
5

**Fig 7. Most abundant taxa in microbialites of Quintana Roo.** The heatmap shows the 25 most abundant across sites.

lake (8.5 and 5.9%, respectively) compared to Chichancanab lake and Cenote Azul (0.6 and 1.5%). Something similar occurred with the genus SBR1031 of the same phylum. Some differences were more noticeable than others; for instance, the prevalence of the genus *Pseudomonas* was the highest at Bacalar lake (South) (25.8%) compared with the rest of the sampling points, and these taxa were not found at all in Cenote Azul. Genus *Aeromonas, Stenotrophomonas* and *Brevundimonas* were more prevalent in Bacalar lake (South). *Bacillus* was highest in Bacalar lake (South) (8.8) and absent in Cenote Azul and Azul lake. Another noticeable difference occurred with the prevalence of *Rhodomicrobium* genus, which was more prevalent in Chichancanab lake (11.2%) than in any other lake. *Rivularia*_PCC-7116 was more prevalent in Azul lake (8.7%) compared to Muyil lake (2), lower in Bacalar lake (North) (1.3%) and not present in Cenote Azul. Microsillaceae and an unknown family of Pseudomonadota were more prevalent in Muyil lake (5.5 and 5.8%, respectively) compared to other sites and non-existent in Cenote Azul. Taxa that were more prevalent in Cenote Azul compared with other lakes include genera NB1-j, the Cyanobacteriota family Mycrocistaceae, Nitrosopumilaceae of Crenarchaeota, genus OM190 of Planctomycetota and *Nitrospira*.

## Discussion

### Characteristics of southeastern Yucatan Peninsula as a hotspot zone for microbialites

The Yucatan Peninsula stands out as a climatically dynamic region, where its geological history and hydrogeochemical characteristics have played a crucial role in shaping a karst landscape that harbors diverse terrestrial and aquatic ecosystems. In its southeastern region, freshwater exhibits sulfate, bicarbonate, calcium, and magnesium ions saturation. The interaction of these ions with microbial communities and factors such as water quality, transparency, and temperature, among other aspects, has led to the formation of a hotspot region for the occurrence of microbialites. The Yucatan Peninsula's contemporary landscape has been configured by the impact of the Chicxulub asteroid 65 million years ago. In addition, it has given rise to two crucial ecosystems within the karstic environment: sinkholes (cenotes, both caverns and superficial) and lakes (formed by geological faults). Both freshwater systems are intricately connected and primarily fed by underground water. Landscape modifications in the Yucatan Peninsula, extending from the Pleistocene to the present, have also been significantly influenced by fluctuations in mean sea level during the last glaciation [49]. Researchers in the Yucatan Peninsula commonly rely on sediment and stalagmite samples as proxies to palaeoecological studies [50,51].

Freshwater ecosystems in southeastern Yucatan Peninsula are characterized as oligotrophic, marked by low levels of phosphorus and nitrogen, which are common traits in many freshwater sites conducive to the prospering of microbialites [52]. The "hard water" characteristics, due to the saturation of ions such as sulfates, bicarbonates, calcium, and magnesium, are a positive sign for the occurrence of microbialites. Localities analyzed here do not present problems of nutrient increase, except for Bacalar lake (South). The presence of ammonium and NOx in Bacalar lake (South) has been previously documented [12], with these ions associated with anthropogenic activities including deforestation, intensive agriculture and insufficient sewage treatment [53]. The hydrogeochemistry variables of Bacalar lake (North) appeared to be linked to its connectivity with Chile Verde lake, Salada lake, and the Chetumal Bay estuary [15].

Conversely, Bacalar lake (South) features cenotes within the lake, contributing to bicarbonate saturation [54]. Despite their geographical proximity, Chichancanab lake and Azul lake exhibited variations in the concentrations of dissolved ions in their waters. The saturation of sulfate ions in the water of Chichancanab lake is associated with the dissolution of gypsum/anhydrite bedrock in the basin, possibly due to the Chicxulub impact [29]. For Azul lake, this study represents the first report on the presence of microbialites. It provides a complete mineralogical, structural, geochemical, and hydrogeochemical characterization of the site (S6 Table). The hydrogeochemical characteristics of Muyil lake are probably associated with its proximity to the Caribbean Sea (Fig 1).

The mineralogy of the microbialites in the Yucatan Peninsula reveals different forms of carbonates, with calcite being the predominant mineral (> 94%) in the microbialites of the southern region. Chichancanab lake and Azul lake exhibited

microbialites composed primarily of magnesian calcite, constituting over 98% of their composition, with traces (<2%) of aragonite. This mineral has also been documented in microbialites, such as those found in Alchichica crater-lake in central Mexico and Dziani Dzaha in the Indian Ocean [55,56]. In the case of Muyil, a coastal lake, calcite dominated its structure (60%), while gypsum constituted 38.85%. These percentages closely resemble those reported in microbialites from Cayo Sabinal, Cuba, which also exhibit marine influences [56].

Geochemically, the microbial mat, specifically the microbialite structure, relies on an extracellular polymeric substance (EPS) matrix that undergoes enrichment through adsorption, trapping, and metal binding mechanisms [57]. Essential chemical elements, acting as cofactors, play pivotal roles in various enzymatic metabolic pathways and contribute to the promotion of nutrient cycling. Previous studies on microbialites have documented the presence of calcium, iron, and strontium, establishing them as integral components of Yucatan Peninsula microbialites [56,57]. The trace elements rubidium (Rb) and niobium (Nb) receive particular attention, which have not been extensively explored in microbialites from Quintana Roo. The presence of Rb was reported in water from Bacalar lake [57].

Further exploration into the geochemical behavior of niobium compounds reveals their affinity for organic agents and slight solubility in both acidic and alkaline conditions [58]. Rubidium cation is an understudied alkaline trace element that has been identified in both humans and other organisms [59]. Niobium cation was only described for Chichancanab lake and Cenote Azul, which have a stronger groundwater connectivity [29]. One plausible explanation for its presence in the microbialite system may be attributed to natural runoff during the rainy season, stemming from environmental sources such as rock weathering or coal combustion [60]. It is crucial to highlight the necessity for a comprehensive understanding of these elements' geochemical behavior within the Yucatan Peninsula's natural karst systems. Establishing background values for geochemical elements in soil samples, water, sediment, and microbialites, which are integral components of the watersheds, is imperative. This information is essential for developing effective spatial and temporal monitoring strategies to support water quality care and prevent eutrophication processes.

## Microbialites as bio-sensors of water quality

The emergent properties of microbial communities, including diversity, richness, and dominance, have been identified as biosensors for various environmental perturbations or trophic states in aquatic ecosystems [61]. Increased nutrient levels in the water column can lead to changes in water coloration, acidity, turbidity, and dissolved oxygen, among other factors. These alterations drive shifts in microbial taxonomic profiles as responses to environmental changes, subsequently influencing the health conditions of microbialites [62,63]. Microbial communities associated with microbialites strongly correlate with the trophic state of the water column [12,62]. Lake Bacalar (South) had the lowest diversity and richness metrics among the sites analyzed in this study (Figs 2, 4, and 6). This pattern coincides with a previous report, where the southern region of Lake Bacalar presented dysbiosis, associated with a higher concentration of nitrate and ammonium [12]. Microbial communities associated with microbialites are sensitive to climate change, nutrient enrichment, and human-induced landscape disturbance [62,63]. We recognize that monitoring efforts of microbialite ecosystems are important to safeguard cultural/social aspects, environmental health, as well as taxonomic, genetic and metabolic biodiversity.

In general, microbialites in the Yucatan Peninsula are found in sites with environmental protection categories (such as Muyil lake in the Sian Ka'an Biosphere Reserve). They are also found in areas where tourism and recreational activities do not have a direct impact on water quality (Chichancanab lake, Azul lake, Bacalar lake North and Cenote Azul), resulting in highly diverse microbial communities (Fig 2). Given the importance of sinkholes and lakes in southeastern Yucatan Peninsula, it is imperative to have policies that protect water quality, including sustainable tourism and economic activities that control land use modification, and promote ecosystem adaptation to climate change. These strategies should include nutrient monitoring and microbialite diversity as biosensors of water quality.

## Composition of microbialites

The bacterial composition of shallow microbialites is generally represented by Pseudomonadota, Bacillota, Cyanobacteriota, Bacteroidota, Chloroflexota, and Planctomycetota, as commonly described for both marine and freshwater communities. In the case of deeper environments, such as Cenote Azul, a higher abundance of archaea (Crenarchaeota), Acidobacteriota, and Nitrospirota has been reported [64,65]. Furthermore, locations with microbialite presence offer valuable opportunities for studies aimed to describe the eukaryotic community (including fish, nematodes, isopods, bivalves, and diatoms) that utilize microbialites as habitats or refuges [66,67]. Visually, microbialites exhibit robust connectivity with eukaryotic groups, and future research of the eukaryotic community associated with microbialites will enhance our understanding of their formation processes.

The deep microbialites sampled in the Cenote Azul have a significantly different composition than shallow sampling sites (S3 Fig). Compared to the other lakes, Cenote Azul exhibits the highest values in alpha diversity metrics. Beta diversity analysis further highlights substantial dissimilarities between Cenote Azul and other sites. Additionally, Cenote Azul stands out as the site with the highest presence of archaeal groups (Crenarchaeota), uncultured groups of Planctomycetota, Pseudomonadota and NB1-j phyla, a group of Cyanobacteriota (Mycrocistecea) that is not present in the other microbialites (Fig 7). The distinctiveness of Cenote Azul is further underscored by its contrast with the nearest site, Bacalar lake, which has previously been reported for its unique eukaryotic and prokaryotic groups [27,28,30]. The Cenote Azul is characterized as a less permeable limestone rock system; hence, some authors regard it as a water body with limited connectivity to Bacalar lake [29,30].

Cyanobacteriota and Pseudomonadota (especially Alphaproteobacteria) were the predominant phyla in Yucatan Peninsula microbialites, which, along with Chloroflexota and Chlorobiota, are considered photosynthetic, both oxygenic and anoxygenic, with metabolisms associated with carbonate precipitation [68]. Regarding Alphaproteobacteria, the orders Rhodobacterales and Caulobacterales were the most abundant, respectively. These microbial groups have been implicated in microbialite formation processes, including anoxygenic photosynthesis, extracellular polymeric substance (EPS) degradation, and carbon and nitrogen fixation [56,69].

This study suggests the presence of a new family of Cyanobacteriota, representing 43–50% across the sites of Chichancanab lake, Azul lake, Muyil lake, and Bacalar (North and South). In comparison, it constitutes only 10% in Cenote Azul. Cyanobacteria, both filamentous and unicellular, have been described as playing an essential role in the formation of both deep and shallow microbialites [27,28,65,68]. Cyanobacteriota in microbialites perform various functions that promote microbialite growth, including extracellular polymeric substance (EPS) synthesis, oxygenic photosynthesis, ammonification, sulfur assimilation, carbon dioxide and nitrogen fixation [28]. The most common families of Cyanobacteriota in microbialites were Nostocaceae, Xenococcaceae, Chroococcidiopsaceae, Phormidiaceae, and Gloeocapsaceae, except for microbialites in Cenote Azul where Microcystaceae was dominant. It was interesting to note a high dominance of *Planktothrix* in microbialites from Chichancanab lake, which represented up to 14% of Cyanobacteriota relative abundance. Also, *Planktothrix* was present in low abundance in Muyil lake (4%). *Planktothrix* have been associated with eutrophic systems where they compete with benthic cyanobacteria for light, this planktonic species requires more attention in the Yucatan Peninsula, as they are associated with the production of cyanotoxins harmful to mammals [27,62].

Crenarchaeota is the most abundant archaeal phylum in Yucatan Peninsula microbialites, standing out in the Cenote Azul. Crenarchaeota (mesophyll) and Euryarchaeota are recognized in marine and estuarine anaerobic sediments [70]. Crenarchaeota is a metabolically diverse group, encompassing anaerobic heterotrophic subgroups that employ electron acceptors, including sulfate, for various metabolic processes [71]. Águila et al. [28] report the presence of chemolithotrophic metabolisms in the microbialites of the Cenote Azul. Other archaeal phyla observed include Nanoarchaeota, Thermoplasmatota, Micrarchaeota, Halobacterota, Euyarchaeota, and Asgardarchaeota, among others (S1 Fig). Overall, the diversity of Archaea is closely linked to microorganisms engaged in autotrophy, chemolithotrophy, and nitrification activities

[72,73]. Considering the phosphorus and nitrogen limitations under which microbialites form in southeastern Yucatan Peninsula, future research endeavors must explore diverse metabolisms. This approach will offer valuable insights into the dynamics of microbial communities and their role in the lithification process of microbialites.

In the Upset analysis, we found that only 244 ASVs are shared among all sites (Fig 6). These results suggest that each microbialite site is highly distinct, containing bacteria and archaea that play specific roles in mineral precipitation. Although the process of microbialite formation is not yet fully understood, these sites become model study areas for future research. Using LCBD and SCBD metrics, we identified that the sites experiencing community turnover are mainly Cenote Azul and Bacalar lake (South) (S4 and S5 Table). Bacterial genera belonging to Pseudomonadota and Cyanobacteriota were the main contributors to changes in the structures of microbial assemblages associated with microbialites, with notable mentions including *Rhodomicrobium*, *Rivularia* PCC-7116, *Pseudomonas* (S5 Table), groups strongly associated with the formation of carbonate microbialites [74].

Together, these results highlight that differentiated communities maintain similar functional groups, suggesting that the emergent properties of microbialite-associated communities respond to local environmental changes that are directly influenced by anthropogenic factors. Significant attention should be given to the considerable presence of uncultured microorganisms within the freshwater microbialites of Quintana Roo. Future metagenomic approaches stand as a viable means to unravel the identity of these novel species and understand the metabolic pathways that contribute to the process of lithification.

## Conclusions

The physicochemical conditions (temperature, pH, solar irradiation, sulfate saturation, bicarbonate, calcium, and magnesium ions) of surface and deep aquatic environments in southeastern Yucatan Peninsula make it a hotspot for microbialite occurrences. Other regions with high microbialite diversity include Hamelin Pool, Shark Bay, Western Australia [75], the Central Andes region [76], crater lakes of the Trans-Mexican Volcanic Belt [73] and Cuatro Ciénegas, Mexico [77]. Mexico bears a significant responsibility as one of the countries with the highest presence of sites hosting microbialites. The conservation of microbialites may be threatened by climate change-related situations such as intense droughts and rains. Currently, the most detrimental activities are anthropogenic at the local level, often causing irreversible damage, as seen in the extraction of groundwater in Cuatro Ciénegas, Coahuila [78,79]. It is crucial to recognize the importance of microbes, including microbialites, and understand their ecological role. Our investigation brings attention to the importance of the high connectivity between aquatic systems in the karstic environments of the Yucatan Peninsula. Caring for microbialites involves watershed management, with primary attention to land use changes and water quality.

## Supporting information

**S1 Fig. Relative abundances of the different Archaea phyla (a) and family (b) on the microbialites from Quintana Roo lakes.**
(TIF)

**S2 Fig. Heat map of the 15 most abundant families of the Bacterial community; values are presented in percentages (%).**
(TIF)

**S3 Fig. Bacterial community composition of microbialites samples as according to NMDS ordination comparing shallow and deep categories.**
(TIF)

**S1 Table. Location and characteristics of each study site.**
(DOCX)

**S2 Table. Alpha diversity indexes of microbial communities associated with microbialites in the lakes of Quintana Roo.**
(DOCX)

**S3 Table. Permutational multivariate analysis of variance (PERMANOVA) of microbial communities associated with microbialites (a); between shallow and deep sites (b).**
(DOCX)

**S4 Table. LCBD *p-values* of the sites with microbialites in Quintana Roo.**
(DOCX)

**S5 Table. Main species that contribute to the turnover of prokaryotic communities in microbialites of Quintana Roo.**
(DOCX)

**S6 Table. Ion abundance in water column for each study site harboring microbialites in Quintana Roo.**
(DOCX)

## Acknowledgments

Technical assistance is acknowledged to O Gaona (I Ecología, UNAM), T Pi and O Zamora (I Geología, UNAM).

## Author contributions

**Conceptualization:** Alfredo Yanez-Montalvo, Arturo Bayona, Luisa I. Falcón.

**Data curation:** Alfredo Yanez-Montalvo, Bernardo Águila, Arit S de León-Lorenzana, Luisa I. Falcón.

**Formal analysis:** Alfredo Yanez-Montalvo, Bernardo Águila, Arit S de León-Lorenzana, Pavel Popoca, Luisa I. Falcón.

**Funding acquisition:** Luisa I. Falcón.

**Investigation:** Alfredo Yanez-Montalvo, Bernardo Águila, Arit S de León-Lorenzana, Arturo Bayona, Nuria Torrescano-Valle, Luisa I. Falcón.

**Methodology:** Alfredo Yanez-Montalvo, Arit S de León-Lorenzana, Arturo Bayona, Nuria Torrescano-Valle, Pavel Popoca, Luisa I. Falcón.

**Project administration:** Luisa I. Falcón.

**Resources:** Luisa I. Falcón.

**Software:** Alfredo Yanez-Montalvo, Bernardo Águila, Luisa I. Falcón.

**Supervision:** Luisa I. Falcón.

**Validation:** Alfredo Yanez-Montalvo, Bernardo Águila, Arit S de León-Lorenzana, Luisa I. Falcón.

**Visualization:** Alfredo Yanez-Montalvo, Arit S de León-Lorenzana, Nuria Torrescano-Valle, Pavel Popoca, Luisa I. Falcón.

**Writing – original draft:** Alfredo Yanez-Montalvo, Bernardo Águila, Arit S de León-Lorenzana, Arturo Bayona, Nuria Torrescano-Valle, Pavel Popoca, Luisa I. Falcón.

**Writing – review & editing:** Alfredo Yanez-Montalvo, Bernardo Águila, Arit S de León-Lorenzana, Arturo Bayona, Nuria Torrescano-Valle, Pavel Popoca, Luisa I. Falcón.

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
