## [Decision Letter · Decision Letter 0]

16 Jul 2024

PONE-D-24-25938Karst-environments of the southeastern Yucatan Peninsula: hotspots for modern freshwater microbialitesPLOS ONE

Dear Dr. Falcón,

Thank you for submitting your manuscript to PLOS ONE. After careful consideration, we feel that it has merit but does not fully meet PLOS ONE’s publication criteria as it currently stands. Therefore, we invite you to submit a revised version of the manuscript that addresses the points raised during the review process.

We look forward to receiving your revised manuscript.

Kind regards,

Yizhi Sheng

Academic Editor

PLOS ONE

Journal Requirements:

When submitting your revision, we need you to address these additional requirements. 1. Please ensure that your manuscript meets PLOS ONE's style requirements, including those for file naming. The PLOS ONE style templates can be found at  https://journals.plos.org/plosone/s/file?id=wjVg/PLOSOne_formatting_sample_main_body.pdf and https://journals.plos.org/plosone/s/file?id=ba62/PLOSOne_formatting_sample_title_authors_affiliations.pdf 2. We note that the grant information you provided in the ‘Funding Information’ and ‘Financial Disclosure’ sections do not match.  When you resubmit, please ensure that you provide the correct grant numbers for the awards you received for your study in the ‘Funding Information’ section. 3. Thank you for stating the following financial disclosure:  "This work had funding from UNAM, PAPIIT (IN204224) to LIF. AY and BA hold postdoc fellowships from CONAHCyT and UNAM-PAPIIT, respectively." Please state what role the funders took in the study. If the funders had no role, please state: "The funders had no role in study design, data collection and analysis, decision to publish, or preparation of the manuscript." If this statement is not correct you must amend it as needed.  Please include this amended Role of Funder statement in your cover letter; we will change the online submission form on your behalf. 4. Thank you for stating the following in the Acknowledgments Section of your manuscript:  "This work had funding from UNAM, PAPIIT (IN204224) to LIF. AY and BA hold postdoc fellowships from CONAHCyT and UNAM-PAPIIT, respectively. Technical assistance is acknowledged to O Gaona (I Ecología, UNAM), T Pi and O Zamora (I Geología, UNAM)." However, please note that funding information should not appear in the Acknowledgments section or other areas of your manuscript. We will only publish funding information present in the Funding Statement section of the online submission form. Please remove any funding-related text from the manuscript. 5. When completing the data availability statement of the submission form, you indicated that you will make your data available on acceptance. We strongly recommend all authors decide on a data sharing plan before acceptance, as the process can be lengthy and hold up publication timelines. Please note that, though access restrictions are acceptable now, your entire data will need to be made freely accessible if your manuscript is accepted for publication. This policy applies to all data except where public deposition would breach compliance with the protocol approved by your research ethics board. If you are unable to adhere to our open data policy, please kindly revise your statement to explain your reasoning and we will seek the editor's input on an exemption. Please be assured that, once you have provided your new statement, the assessment of your exemption will not hold up the peer review process.  6. We note that Figure 1 in your submission contain map images which may be copyrighted. All PLOS content is published under the Creative Commons Attribution License (CC BY 4.0), which means that the manuscript, images, and Supporting Information files will be freely available online, and any third party is permitted to access, download, copy, distribute, and use these materials in any way, even commercially, with proper attribution. For these reasons, we cannot publish previously copyrighted maps or satellite images created using proprietary data, such as Google software (Google Maps, Street View, and Earth). For more information, see our copyright guidelines: http://journals.plos.org/plosone/s/licenses-and-copyright. We require you to either (1) present written permission from the copyright holder to publish these figures specifically under the CC BY 4.0 license, or (2) remove the figures from your submission: 1) You may seek permission from the original copyright holder of Figure 1 to publish the content specifically under the CC BY 4.0 license.   We recommend that you contact the original copyright holder with the Content Permission Form (http://journals.plos.org/plosone/s/file?id=7c09/content-permission-form.pdf) and the following text:“I request permission for the open-access journal PLOS ONE to publish XXX under the Creative Commons Attribution License (CCAL) CC BY 4.0 (http://creativecommons.org/licenses/by/4.0/). Please be aware that this license allows unrestricted use and distribution, even commercially, by third parties. Please reply and provide explicit written permission to publish XXX under a CC BY license and complete the attached form.” Please upload the completed Content Permission Form or other proof of granted permissions as an ""Other"" file with your submission. In the figure caption of the copyrighted figure, please include the following text: “Reprinted from [ref] under a CC BY license, with permission from [name of publisher], original copyright [original copyright year].” 2) If you are unable to obtain permission from the original copyright holder to publish these figures under the CC BY 4.0 license or if the copyright holder’s requirements are incompatible with the CC BY 4.0 license, please either i) remove the figure or ii) supply a replacement figure that complies with the CC BY 4.0 license. Please check copyright information on all replacement figures and update the figure caption with source information. If applicable, please specify in the figure caption text when a figure is similar but not identical to the original image and is therefore for illustrative purposes only. The following resources for replacing copyrighted map figures may be helpful: USGS National Map Viewer (public domain): http://viewer.nationalmap.gov/viewer/The Gateway to Astronaut Photography of Earth (public domain): http://eol.jsc.nasa.gov/sseop/clickmap/Maps at the CIA (public domain): https://www.cia.gov/library/publications/the-world-factbook/index.html and https://www.cia.gov/library/publications/cia-maps-publications/index.htmlNASA Earth Observatory (public domain): http://earthobservatory.nasa.gov/Landsat:
http://landsat.visibleearth.nasa.gov/USGS EROS (Earth Resources Observatory and Science (EROS) Center) (public domain): http://eros.usgs.gov/#Natural Earth (public domain): http://www.naturalearthdata.com/

Reviewers' comments:

Reviewer's Responses to Questions

**Comments to the Author**

1. Is the manuscript technically sound, and do the data support the conclusions?

Reviewer #1: Yes

Reviewer #2: Partly

2. Has the statistical analysis been performed appropriately and rigorously? 

Reviewer #1: Yes

Reviewer #2: Yes

3. Have the authors made all data underlying the findings in their manuscript fully available?

Reviewer #1: Yes

Reviewer #2: No

4. Is the manuscript presented in an intelligible fashion and written in standard English?

Reviewer #1: Yes

Reviewer #2: Yes

5. Review Comments to the Author

Reviewer #1: The manuscript titled "High Connectivity of Microbial Communities in Karst Aquifers: A Study of Microbialites in the Yucatan Peninsula" presents a comprehensive study on the hydrogeochemical profile, mineralogy, elemental geochemistry, and microbial composition of microbialites from the Yucatan Peninsula. With some minor revisions to improve clarity and readability, this paper should make a significant contribution to the field of microbialite ecology and conservation. The authors should address the grammatical issues, consider revising some figures for clarity, and potentially expand on the implications of their findings for environmental monitoring and conservation efforts.

Ln 61-66 This sentence is too long. Consider to break it down.

Ln 141 Remove the comma “.”

Ln 174 hclust function?

Ln 189 Do not italic, keep the format consistent throughout

Provide high-resolution figures (e.g., figure 7) for better readability.

Ln 257 Is that common to see high levels of rubidium and niobium in the lake that forms microbialites? What are the potential implications of these trace elements for microbialite formation or ecology?

The manuscript suggested that microbialites could serve as biosensors for water quality. Could the authors elaborate on how this might be implemented in practice? What specific microbial community changes would indicate declining water quality?

The authors mentioned the need for conservation of microbialites. Based on their findings, what specific conservation measures would they recommend for the microbialite sites in the Yucatan Peninsula?

The study also mentioned the presence of eukaryotic organisms associated with microbialites. I wonder how might these eukaryotes influence microbialite formations and their stabilities.

Reviewer #2: This work is a significant contribution, and it should be published. However, as a comparative study, the authors have overlooked several facts occurring in this region. Some of the results from this study do not compare well with previous results published by them or other researchers. I will present a more detailed account of them in the following paragraphs.

Additionally, I detect some errors in their interpretations and affirmations. This region is unique in the world, and all papers published should establish a good comprehensive understanding of existing literature together with the results presented in this manuscript, to help in more integral actions for their conservation, not only for the microbialites, because they are part of complex and fragile ecosystems with many interactions not known. The region has been seriously affected by a disordered growth of population and tourism and, lately by a controversial train:

Martínez-Romero, E., G. Gasparello & M. A. Díaz-Perera, 2023. Territorios mayas en el paso del tren. Riesgos previsibles y posturas independientes sobre el tren maya. Bajo Tierra Ediciones, Mexico. 332 p.

Ortega, R. P. & I. G. Jaber, 2022. ENVIRONMENT A controversial train heads for the Maya forest. Science 375(6578):250-251.

The only way I envisage is using the data presented in the literature to support a better decision from the whole society to conserve this fragile region.

It's crucial to clearly distinguish between new and previously reported data to avoid confusion. For instance, in lines 521-524, the authors discuss the dominant species in Cenote Azul, but they reported in their citation [27] that 70% was Microcystaceae from Cenote Azul. They do not cite their previous study. This lack of distinction is not unique to this instance, making it difficult to discern new information from previously reported data in the whole paper. Accurate citation is a fundamental part of our academic community, and it's essential that we all play our part in maintaining its integrity. Additionally, Planktothrix is not mentioned in the citation [27] (see Line 527), at least not in my reading.

Some punctual remarks are as follows:

Line 63.- It is not the second, it is the deepest:

Alcerreca-Huerta, J. C., O. F. Reyes-Mendoza, J. A. Sanchez-Sanchez, T. Alvarez-Legorreta & L. Carrillo, 2024. Recent records of thermohaline profiles and water depth in the Taam ja’ Blue Hole (ChetumalBay, Mexico). Frontiers in Marine Science 11:1387235. doi:10.3389/fmars.2024.1387235.

Line 64.- What about Calakmul reserve, it is bigger than Sian Ka’an.

Line 65.- The authors should have mentioned Chichancanab in the introduction, where a unique flock of Cyprinodon speciates. It is the second largest lake from the Yucatan Peninsula and hosts microbialites, as they will report later in this manuscript. Moreover, it is part of the systems studied here, but at this point of the manuscript, I did not detect it.

Strecker, U. S. u.-h., de), 2004. The Cyprinodon species flock from Laguna Chichancanab, Mexico (Teleostei): sexual and

disruptive selection driving adaptive radiation. Mitteilungen aus dem Hamburgischen Zoologischen Museum und Institut 101:65-74.

In Fig. 1 authors should explain the studied sites with detail.

Line 97.- Please provide coordinates of the sampling sites; microbialites are not the same in large lakes as in Bacalar. Later, the authors said they sampled Bacalar North and South. Besides, the results presented in previous studies must be contrasted with those of this new study.

In addition, how did they sample five sites in small and deep systems such as Cenote Azul and large systems such as Bacalar, Chichancanab, or Muyil? At what deep? Cenote Azul is 65 m deep, and Bacalar averages 8.5 m. Moreover, on which dates did the authors take the sampling? A severe brownification event in Bacalar in 2020 lasted more than one year for a full recovery of the system. We do not know the effect of this event on the stromatolites:

Carrillo, L., M. Yescas, M. O. Nieto-Oropeza, M. Elías-Gutiérrez, J. C. Alcérreca-Huerta, E. Palacios-Hernández & O. F. Reyes-Mendoza, 2024. Investigating the Morphometry and Hydrometeorological Variability of a Fragile Tropical Karstic Lake of the Yucatán Peninsula: Bacalar Lagoon. Hydrology 11(68) doi:10.3390/hydrology11050068.

Line 105.- Maybe the largest in the world:

Gischler, E., M. A. Gibson & W. Oschmann, 2008. Giant Holocene freshwater microbialites, Laguna Bacalar, Quintana Roo, Mexico. Sedimentology 55(5):1293-1309.

Line 109.- It was a personal observation or it was published?

Perry, E., G. Velazquez-Oliman & L. Marin, 2002. The hydrogeochemistry of the karst aquifer system of the northern Yucatan Peninsula, Mexico. International Geology Review 44(3):191-221.

Line 110.- The maximum depth reported and detected is 65 m, not 100 m. How the authors find stromatolites at 70 m? In ther citation 27, same authors collected up to 30 m, and they do not report to 70 m, as well the citation number 30 does not say anything about stromatolites to 70 m. The most interesting point in the citation 30 is that Cenote Azul has almost an entirely different composition of zooplankton than Bacalar. Then, not only the water chemistry is different, also the communities, including the stromatolites.

Cervantes-Martínez, A., M. Mezeta-Barrera & M. Gutiérrez-Aguirre, 2009. Basic limnology of the karstic tourist lake Cenote Azul in Quintana Roo, Mexico. Hidrobiologica 19(2):177-180.

Line 114-122.- Apart of the points remarked by authors Chichancanab hosts a speciation of Cyprinodon (from marine origin) quite special, and they do not mention it. I mentioned this point previously.

Line 123-131.- Here the authors do not mention a comparison of Muyil with other lakes nearby

Valdez-Moreno, M., M. Mendoza-Carranza, E. Rendon-Hernandez, E. Alarcon-Chavira & M. Elias-Gutierrez, 2021. DNA Barcodes Applied to a Rapid Baseline Construction in Biodiversity Monitoring for the Conservation of Aquatic Ecosystems in the Sian Ka'an Reserve (Mexico) and Adjacent Areas. Diversity-Basel 13(7):22 doi:10.3390/d13070292.

Line 234.- Muyil lake is about 9 km from the shoreline, and is connected indirectly to the sea by a complex series of channels. I believe is not extrictely “coastal”. It does not show brackish conditions as any coastal lake.

Line 237.- NOx and ammonium values are too high. Did the authors some crosslab comparisons?

Line 239.- The authors should have mentioned that they sampled Bacalar North and South in the methods section. That is why they should detail how the sampling was conducted and on which dates. In their previous cited paper [12], they found differences between the north and south of Bacalar.

Line 316-317.- There is no explanation of sampling deeps for Cenote Azul in methods section.

Line 334-336.- The authors did not say the deeps of their samplings. Moreover, they found differences in the different deeps of Cenote Azul in a previous study (see reference 27).

Line 404.- The values presented in the previous work are quite different as far as I can see. That is the reason to know dates of sampling.

Line 405.- These systems are connected through a complex series of channels, but how they are connected should be considered: with Chile Verde, this connection is on the north side of Bacalar.

Line 417-418.- What about the fauna? There is some penetration of the marine fauna? It is documented.

Line 441.- I do not agree. Most (if not all) water inflows to Bacalar are also groundwaters. Some superficial sporadic inflows are present in rain season in the north.

Line 456-463.- Because of these reasons, it is important to know the sampling dates. Bacalar Lake and its associated systems passed an intense event of brownification and it should be established a before-after baseline.

It apparently caused also mortality in several of the communities from this site, for example (unfortunately, this report is gray literature):

Castro-Chan, R.A.; De Jesús Navarrete, A.; Zavala Mendoza, A. Estudio Para Establecer Las Causas de La Mortalidad Masiva Del Caracol de Agua Dulce Chivita Pomacea Flagellata; Consejo Quintanarroense de Ciencia y Tecnología: Chetumal, Mexico, 2021.

Line 492-497.- It is unclear if these are results from this study or the previous study because the authors do not cite it in this part, and the sampling they used is unclear. See citation 27.

Line 518.- What about citation 27, it compares different depths in Cenote Azul.

Line 530.- Please add some citations.

Line 562-565.- It is not at all accurate; the communities are realizing the relevance of microbialites, and some examples exist (maybe small or limited, but they are starting points):

O'Kelard, A., 2021. Don Lito habla desde Bacalar: Importancia de los estromatolitos en el origen de la vida en el planeta. Global Design. 121 p.

Line 582-585. The authors miss out on the fact that each system studied seems unique, with a particular bacteria association. The stromatolites are part of the whole picture of each system's biodiversity. Then, by the end of the study, it is unclear whether connectivity exists. As far as I can understand, all studied systems are different in composition in microbialites and turnover, with different chemistry of the waters. These results can be obvious if comparing, for example, Cenote Azul and Bacalar, which are about 200 m apart but also occur in more or less similar physiographically systems such as Chichancanab and Muyil or Bacalar.

Although the manuscript's scope is to highlight the Yucatan Peninsula as a hotspot for modern microbialites, the authors should mention the 2020 brownification event and the posterior recovery of some of these systems. They should also mention the actual challenges and dangers for these ecosystems. If the anthropogenic effects continue, we are approaching a non-returning point, with a possible severe consequence for the stromatolites and the other communities dwelling in these unique aquatic ecosystems.

6. PLOS authors have the option to publish the peer review history of their article (what does this mean? ). If published, this will include your full peer review and any attached files.

**Do you want your identity to be public for this peer review?** For information about this choice, including consent withdrawal, please see our Privacy Policy .

Reviewer #1: No

Reviewer #2: No

---

## [Author Response · Author response to Decision Letter 0]

28 Jan 2025

Reviewer #1: The manuscript titled "High Connectivity of Microbial Communities in Karst Aquifers: A Study of Microbialites in the Yucatan Peninsula" presents a comprehensive study on the hydrogeochemical profile, mineralogy, elemental geochemistry, and microbial composition of microbialites from the Yucatan Peninsula. With some minor revisions to improve clarity and readability, this paper should make a significant contribution to the field of microbialite ecology and conservation. The authors should address the grammatical issues, consider revising some figures for clarity, and potentially expand on the implications of their findings for environmental monitoring and conservation efforts.

Ln 61-66 This sentence is too long. Consider to break it down.

A= Thank you for your comment. The suggestion was made.

Ln 141 Remove the comma “.”

A= Thank you for your comment. The suggestion was made.

Ln 174 hclust function?

A= Thank you for your comment. The suggestion was made.

Ln 189 Do not italic, keep the format consistent throughout

A= Thank you for your comment. The suggestion was made.

Ln 257 Is that common to see high levels of rubidium and niobium in the lake that forms microbialites? What are the potential implications of these trace elements for microbialite formation or ecology?

A= There are few studies based on the geochemical characterization of microbialites and their relationship to microbial composition, which we discuss in lines 440-45. In addition, the presence is based on the particularities of each site, for the case of the Quintana Roo´s Lakes, there are no background values of sediment geochemistry.

The manuscript suggested that microbialites could serve as biosensors for water quality. Could the authors elaborate on how this might be implemented in practice? What specific microbial community changes would indicate declining water quality?

A= Emergent properties of microbial communities have been proposed as biosensors for various environments, including coral reefs and the water column (Kiersztyn et al., 2019). We highlight two instances where the microbiomes of microbialites have experienced dysbiosis as a result of human activities (Lindsay et al., 2017; Yanez et al., 2019).

The authors mentioned the need for conservation of microbialites. Based on their findings, what specific conservation measures would they recommend for the microbialite sites in the Yucatan Peninsula?

A= Our working group, along with civil associations and municipal decision-makers, has been actively engaged in science outreach. Additionally, we have promoted the generation of public policies aimed at protecting and conserving the cultural, ecological and evolutionary importance of microbialites in Mexico at the national level.

The study also mentioned the presence of eukaryotic organisms associated with microbialites. I wonder how might these eukaryotes influence microbialite formations and their stabilities.

A=Modern microbialites are home to associated eukaryotic communities. These communities work together to maintain stability within the microbialite ecosystem (Bonacolta et al., 2024). Our group's future research will focus on studying the eukaryotic communities associated with the microbialites at the sites we have examined here.

Reviewer #2: This work is a significant contribution, and it should be published. However, as a comparative study, the authors have overlooked several facts occurring in this region. Some of the results from this study do not compare well with previous results published by them or other researchers. I will present a more detailed account of them in the following paragraphs.

Additionally, I detect some errors in their interpretations and affirmations. This region is unique in the world, and all papers published should establish a good comprehensive understanding of existing literature together with the results presented in this manuscript, to help in more integral actions for their conservation, not only for the microbialites, because they are part of complex and fragile ecosystems with many interactions not known. The region has been seriously affected by a disordered growth of population and tourism and, lately by a controversial train:

Martínez-Romero, E., G. Gasparello & M. A. Díaz-Perera, 2023. Territorios mayas en el paso del tren. Riesgos previsibles y posturas independientes sobre el tren maya. Bajo Tierra Ediciones, Mexico. 332 p.

Ortega, R. P. & I. G. Jaber, 2022. ENVIRONMENT A controversial train heads for the Maya forest. Science 375(6578):250-251.

A=Our group has experience working in this geographical area of Mexico, and we agree with the reviewer about its unique characteristics. We have utilized references to establish a theoretical framework emphasizing the importance of approaching this region from a microbial ecology perspective. While there is a wealth of literature available to support our descriptions, the references we've chosen, along with those suggested by the reviewer, have significantly improved our work. Additionally, it is important to note that our study region faces considerable anthropogenic pressure on its natural ecosystems. We welcome your comments and will incorporate them into our future work. Nonetheless, when Bacalar went brown, after an intense storm event was after these samples were collected. We are working on a time-series analysis of microbialites diversity spanning 2018-present. Here, we will be able to amply discuss how the changes in land cover are affecting aquatic ecosystems in the area, a true catastrophe.

The only way I envisage is using the data presented in the literature to support a better decision from the whole society to conserve this fragile region.

It's crucial to clearly distinguish between new and previously reported data to avoid confusion. For instance, in lines 521-524, the authors discuss the dominant species in Cenote Azul, but they reported in their citation [27] that 70% was Microcystaceae from Cenote Azul. They do not cite their previous study. This lack of distinction is not unique to this instance, making it difficult to discern new information from previously reported data in the whole paper. Accurate citation is a fundamental part of our academic community, and it's essential that we all play our part in maintaining its integrity. Additionally, Planktothrix is not mentioned in the citation [27] (see Line 527), at least not in my reading.

A= We have worked the paper to make it clear that we use the data from previous work in Bacalar and Cenote Azul as a reference for comparison with the new microbialite sites presented in this study.

Some punctual remarks are as follows:

Line 63.- It is not the second, it is the deepest:

Alcerreca-Huerta, J. C., O. F. Reyes-Mendoza, J. A. Sanchez-Sanchez, T. Alvarez-Legorreta & L. Carrillo, 2024. Recent records of thermohaline profiles and water depth in the Taam ja’ Blue Hole (ChetumalBay, Mexico). Frontiers in Marine Science 11:1387235. doi:10.3389/fmars.2024.1387235.

A=Thank you for your comment. We added the information in the text.

Line 64.- What about Calakmul reserve, it is bigger than Sian Ka’an.

A= Thank you for your comment. You are right, Calakmul is bigger than Sian Ka’an. However, we only mentioned sites of the Transversal Coastal Corridor or sites that may influence karst environments of the Yucatan Peninsula where microbialites have been reported.

Line 65.- The authors should have mentioned Chichancanab in the introduction, where a unique flock of Cyprinodon speciates. It is the second largest lake from the Yucatan Peninsula and hosts microbialites, as they will report later in this manuscript. Moreover, it is part of the systems studied here, but at this point of the manuscript, I did not detect it.

Strecker, U. S. u.-h., de), 2004. The Cyprinodon species flock from Laguna Chichancanab, Mexico (Teleostei): sexual and disruptive selection driving adaptive radiation. Mitteilungen aus dem Hamburgischen Zoologischen Museum und Institut 101:65-74.

A= Thank you for your comment. We wanted to enhance the importance of Bacalar, as the site with the largest extension of microbialites. As we mention later Chichancanab lake is a unique site on its own, which we also discussed later in the Materials and Method section.

In Fig. 1 authors should explain the studied sites with detail.

A=Thank you for your comments. We have added more information in the text about the dates of the sampling campaign and the locations in the S1 table.

Line 97.- Please provide coordinates of the sampling sites; microbialites are not the same in large lakes as in Bacalar. Later, the authors said they sampled Bacalar North and South. Besides, the results presented in previous studies must be contrasted with those of this new study.

A=Thank you for your comment. We have added the information in a S1 table. We sampled data from the previously reported work. We have strengthened the discussion to make the contrast with the new study clearer.

In addition, how did they sample five sites in small and deep systems such as Cenote Azul and large systems such as Bacalar, Chichancanab, or Muyil? At what deep? Cenote Azul is 65 m deep, and Bacalar averages 8.5 m. Moreover, on which dates did the authors take the sampling? A severe brownification event in Bacalar in 2020 lasted more than one year for a full recovery of the system. We do not know the effect of this event on the stromatolites:

Carrillo, L., M. Yescas, M. O. Nieto-Oropeza, M. Elías-Gutiérrez, J. C. Alcérreca-Huerta, E. Palacios-Hernández & O. F. Reyes-Mendoza, 2024. Investigating the Morphometry and Hydrometeorological Variability of a Fragile Tropical Karstic Lake of the Yucatán Peninsula: Bacalar Lagoon. Hydrology 11(68) doi:10.3390/hydrology11050068.

A= Thank you for your comment. We are aware of the suggested work. Also, we are aware of the 2020 event. Through a monitoring effort we are analyzing a time series that will include these events.

Line 105.- Maybe the largest in the world:

Gischler, E., M. A. Gibson & W. Oschmann, 2008. Giant Holocene freshwater microbialites, Laguna Bacalar, Quintana Roo, Mexico. Sedimentology 55(5):1293-1309.

A=Thank you for your comment. We have taken it into consideration.

Line 109.- It was a personal observation or it was published?

Perry, E., G. Velazquez-Oliman & L. Marin, 2002. The hydrogeochemistry of the karst aquifer system of the northern Yucatan Peninsula, Mexico. International Geology Review 44(3):191-221.

A=Thank you for your comment. We have taken it into consideration.

Line 110.- The maximum depth reported and detected is 65 m, not 100 m. How the authors find stromatolites at 70 m? In ther citation 27, same authors collected up to 30 m, and they do not report to 70 m, as well the citation number 30 does not say anything about stromatolites to 70 m. The most interesting point in the citation 30 is that Cenote Azul has almost an entirely different composition of zooplankton than Bacalar. Then, not only the water chemistry is different, also the communities, including the stromatolites.

Cervantes-Martínez, A., M. Mezeta-Barrera & M. Gutiérrez-Aguirre, 2009. Basic limnology of the karstic tourist lake Cenote Azul in Quintana Roo, Mexico. Hidrobiologica 19(2):177-180.

A= Thank you for your observation. Indeed, while Perry et al. (2002) report Cenote Azul as having a depth of at least 65 meters, Tobón-Velázquez et al. (2011) reference Perry et al. (2009) and cite the cenote as 90 meters deep. However, Perry et al. (2009) actually report a maximum depth of 64 meters. We are notifying Tobón-Velázquez et al. of this discrepancy for clarification.

Line 114-122.- Apart of the points remarked by authors Chichancanab hosts a speciation of Cyprinodon (from marine origin) quite special, and they do not mention it. I mentioned this point previously.

A= Thank you for your comment. We only mentioned generalities about the Chichankanab lake and mentioned an example with the cited article about its salinity. We are aware of different works that have been done in the Chichankanab Lagoon, but we will address the major literature generated for this lake in a future work in preparation.

Line 123-131.- Here the authors do not mention a comparison of Muyil with other lakes nearby

Valdez-Moreno, M., M. Mendoza-Carranza, E. Rendon-Hernandez, E. Alarcon-Chavira & M. Elias-Gutierrez, 2021. DNA Barcodes Applied to a Rapid Baseline Construction in Biodiversity Monitoring for the Conservation of Aquatic Ecosystems in the Sian Ka'an Reserve (Mexico) and Adjacent Areas. Diversity-Basel 13(7):22 doi:10.3390/d13070292.

A=We appreciate the comment however we were not comparing at that point. We described the lakes in a general way later in the article. Based on the results, we addressed the differences in the discussion.

Line 234.- Muyil lake is about 9 km from the shoreline, and is connected indirectly to the sea by a complex series of channels. I believe is not extrictely “coastal”. It does not show brackish conditions as any coastal lake.

A= Thank you for your comments. We have deleted the words in the text when referring to Muyil Lake.

Line 237.- NOx and ammonium values are too high. Did the authors some crosslab comparisons?

A=Thank you for your comments. We did not make comparisons with other laboratories for this study. However, the information obtained came from university and research center laboratories that have certified methods.

Line 239.- The authors should have mentioned that they sampled Bacalar North and South in the methods section. That is why they should detail how the sampling was conducted and on which dates. In their previous cited paper [12], they found differences between the north and south of Bacalar.

A= Thank you for your comments. That is correct; we have now added this data in the Materials and Methods/Study with columns for sampling date, as well as a detailed sample description in the method section for more clarity.

Line 316-317.- There is no explanation of sampling deeps for Cenote Azul in methods section.

A= Thank you for your comments. Sampling depths are now explained in the methods section; for this work, we grouped all four depths: 5, 10, 20, and 30 meters.

Line 334-336.- The authors did not say the deeps of their samplings. Moreover, they found differences in the different deeps of Cenote Azul in a previous study (see reference 27).

A= Thank you for your comments. Depths are now stated and explained in the method section.

Line 404.- The values presented in the previous work are quite different as far as I can see. That is the reason to know dates of sampling.

A=Thank you for your comments. We mention in the Materials and Methods/Study site and sample collection section the years in which the sampling campaigns of the sites in Figure 1 were carried out.

Line 405.- These systems are connected through a complex series of channels, but how they are connected should be considered: with Chile Verde, this connection is on the north side of Bacalar.

A= Thank you for your comments. Yes, we are aware of the high connectivity in the Bacalar lagoon sites and we mentioned the water flow in the north side of Bacalar lagoon in Line 407.

Line 417-418.- What about the fauna? There is some penetration of the marine fauna? It is documented.

A= Thanks for the suggestion, marine fauna change is documented, yet this study focuses on microbial ecology.

Line 441.- I do not agree. Most (if not all) water inflows to Bacalar are also groundwaters. Some superficial sporadic inflows are present in rain season in the north.

A= Thank you for your comments. We have changed the word influences to connectivity, based on Perry et al. (2012).

Line 456-463.- Because of these reasons, it is important to know the sampling dates. Bacalar Lake and its associated systems passed an intense event of brownification and it should be established a before-after baseline. It apparently caused also mortality in several of the communities from this site, for example (unfortunately, this report is gray literature):

Castro-Chan, R.A.; De Jesús Navarrete, A.; Zavala Mendoza, A. Estudio Para Establecer Las Causas de La Mortalidad Masiva Del Caracol de Agua Dulce Chivita Pomacea Flagellata;

---

## [Decision Letter · Decision Letter 1]

22 Feb 2025

PONE-D-24-25938R1Karst-environments of the southeastern Yucatan Peninsula: hotspots for modern freshwater microbialitesPLOS ONE

Dear Dr. Falcón,

Thank you for submitting your manuscript to PLOS ONE. After careful consideration, we feel that it has merit but does not fully meet PLOS ONE’s publication criteria as it currently stands. Therefore, we invite you to submit a revised version of the manuscript that addresses the points raised during the review process.

We look forward to receiving your revised manuscript.

Kind regards,

Yizhi Sheng

Academic Editor

PLOS ONE

Journal Requirements:

Reviewers' comments:

Reviewer's Responses to Questions

**Comments to the Author**

1. If the authors have adequately addressed your comments raised in a previous round of review and you feel that this manuscript is now acceptable for publication, you may indicate that here to bypass the “Comments to the Author” section, enter your conflict of interest statement in the “Confidential to Editor” section, and submit your "Accept" recommendation.

Reviewer #1: (No Response)

Reviewer #2: All comments have been addressed

2. Is the manuscript technically sound, and do the data support the conclusions?

Reviewer #1: (No Response)

Reviewer #2: Yes

3. Has the statistical analysis been performed appropriately and rigorously? 

Reviewer #1: (No Response)

Reviewer #2: Yes

4. Have the authors made all data underlying the findings in their manuscript fully available?

Reviewer #1: (No Response)

Reviewer #2: Yes

5. Is the manuscript presented in an intelligible fashion and written in standard English?

Reviewer #1: (No Response)

Reviewer #2: Yes

6. Review Comments to the Author

Reviewer #1: The authors addressed all questions and concerns, and I do not have additional comments on this manuscript.

Reviewer #2: The document was much improved, and I think it can be published. It is an important contribution to the knowledge of the microbialites in this region, highlighting its importance. Some minor corrections are needed for the style. I suggest an English native speaker for it.

The only point I want to make is that some conclusions are not based on the results presented in this study, for example, lines 557-564. They can be part of the introduction. Line 568 is an important point, but to my mind, it should be focused differently and based on this study's findings. Lines 571-575 also are out of the results presented here.

Throughout the manuscript, I have only some small observations.

Line 100.- In which season of the year were sampled Muyil, Chichankanaab, and Azul?

Line 391.- Citation 52 is correct for this paragraph?

Line 416-417.- Please give a citation for Alchichica Lake.

7. PLOS authors have the option to publish the peer review history of their article (what does this mean? ). If published, this will include your full peer review and any attached files.

**Do you want your identity to be public for this peer review?** For information about this choice, including consent withdrawal, please see our Privacy Policy .

Reviewer #1: No

Reviewer #2: No

---

## [Author Response · Author response to Decision Letter 1]

21 Mar 2025

PONE-D-24-25938

Karst-environments of the southeastern Yucatan Peninsula: hotspots for modern freshwater microbialites

PLOS ONE

We hereby reply to questions raised to our manuscript, and attach a version showing the required changes, as well as a version with all suggestions accepted.

Best regards,

Luisa I Falcón

Reviewer #2: The document was much improved, and I think it can be published. It is an important contribution to the knowledge of the microbialites in this region, highlighting its importance. Some minor corrections are needed for the style. I suggest an English native speaker for it.

Thank you- we have reviewed the grammar and wording accordingly.

The only point I want to make is that some conclusions are not based on the results presented in this study, for example, lines 557-564. They can be part of the introduction.

Thank you for your suggestion. We have now moved this paragraph to the introduction (lines 76-83).

Line 568 is an important point, but to my mind, it should be focused differently and based on this study's findings. Lines 571-575 also are out of the results presented here.

Thank you. We have modified the text accordingly (lines 564-570).

Throughout the manuscript, I have only some small observations.

Line 100.- In which season of the year were sampled Muyil, Chichankanaab, and Azul?

In the spring of 2020 (lines 110)

Line 391.- Citation 52 is correct for this paragraph?

No, you are right- we have removed from text.

Line 416-417.- Please give a citation for Alchichica Lake.

Yes- citation is Valdespino et al 2018.

---

## [Editor Report · Decision Letter 2]

25 Mar 2025

Karst-environments of the southeastern Yucatan Peninsula: hotspots for modern freshwater microbialites

PONE-D-24-25938R2

Dear Dr. Falcón,

We’re pleased to inform you that your manuscript has been judged scientifically suitable for publication and will be formally accepted for publication once it meets all outstanding technical requirements.

Kind regards,

Yizhi Sheng

Academic Editor

PLOS ONE
---

## [Editor Report · Acceptance letter]

PONE-D-24-25938R2

PLOS ONE

Dear Dr. Falcón,

I'm pleased to inform you that your manuscript has been deemed suitable for publication in PLOS ONE. Congratulations! Your manuscript is now being handed over to our production team.

Kind regards,

on behalf of

Dr. Yizhi Sheng

Academic Editor

PLOS ONE